# Sequence-regulated copolymerization based on periodic covalent positioning of monomers along one-dimensional nanochannels

Shuto Mochizuki[1], Naoki Ogiwara [1], Masayoshi Takayanagi[2,3], Masataka Nagaoka[2,3], Susumu Kitagawa[1,4] & Takashi Uemura [1,3]

The design of monomer sequences in polymers has been a challenging research subject, especially in making vinyl copolymers by free-radical polymerization. Here, we report a strategy to obtain sequence-regulated vinyl copolymers, utilizing the periodic structure of a porous coordination polymer (PCP) as a template. Mixing of $Cu^{2+}$ ion and styrene-3,5-dicarboxylic acid (**S**) produces a PCP, $[Cu(styrene-3,5-dicarboxylate)]_n$, with the styryl groups periodically immobilized along the one-dimensional channels. After the introduction of acrylonitrile (**A**) into the host PCP, radical copolymerization between **A** and the immobilized **S** is performed inside the channel, followed by decomposing the PCP to isolate the resulting copolymer. The predominant repetitive **SAAA** sequence in the copolymer is confirmed by monomer composition, NMR spectroscopy and theoretical calculations. Copolymerization using methyl vinyl ketone also provides the same type of sequence-regulated copolymer, showing that this methodology has a versatility to control the copolymer sequence via transcription of PCP periodicity at the molecular level.

[1] Department of Synthetic Chemistry and Biological Chemistry, Graduate School of Engineering, Kyoto University, Katsura, Nishikyo-ku, Kyoto 615-8510, Japan. [2] Department of Complex Systems Science, Graduate School of Informatics, Nagoya University, Furo-cho, Chikusa-ku, Nagoya 464-8601, Japan. [3] CREST, Japan Science and Technology Agency (JST), 4-1-8 Honcho, Kawaguchi, Saitama 332-0012, Japan. [4] Institute for Integrated Cell-Material Sciences (iCeMS), Kyoto University, Yoshida, Sakyo-ku, Kyoto 606-8501, Japan. Correspondence and requests for materials should be addressed to T.U. (email: uemura@sbchem.kyoto-u.ac.jp)

Natural biopolymers contain perfectly defined monomer sequences in the polymer chain to execute essential biogenic functions. The nucleotide sequence in DNA plays a critical role in genetic inheritance, where the DNA strands with complementary sequences are accurately replicated from the original DNA template within DNA polymerase. According to the nucleotide sequence of the DNA, a peptide chain with a specific structure is produced efficiently via translation of RNA in a ribosome[1]. Thus, nature tells us that the utilization of informative templates in confined spaces would be the key to accomplishing precisely controlled synthesis of polymers with well-defined sequences.

In contrast to the elegant operations of the polymerizations in biological systems, the monomer sequence in a polymer chain is poorly regulated by synthetic chemical processes. Although several approaches with the aid of iterative reactions[2,3], multi-component reactions[4,5], preprogrammed monomers[6,7] and templates[8] have successfully allowed the control of monomer sequences in polymers or oligomers, sequence regulation in vinyl polymers via the chain-growth mechanism has been rather limited. In conventional chain-growth radical polymerization, which is widely used in industry, the composition and sequence distribution of copolymers are governed generally by the inherent relative monomer reactivities, resulting in the production of statistically random copolymers with irregular sequences. For the preparation of vinyl copolymers with regulated sequences, the formation of primitive periodic sequences (XY-, XYY-type sequences) could only be achieved using specific co-monomer pairs[9,10], additives[11,12] and small template molecules[13,14]. Sequential additions of donor and acceptor monomer feeds in living polymerization could virtually allow for controlled placement of the monomers at a chain location[15]. Therefore, sequence-defined radical copolymerization is recognized as the current holy grail of polymer synthesis[16], with strong requirements for protocols that enable the precise positioning of monomers in copolymer chains with high molecular weight[17,18].

Inspired by biological polymerizations, we were drawn to polymer synthesis in confined artificial nanospaces to accomplish the precise control of polymer structures. Porous coordination polymers (PCPs), built up by the self-assembly of metal ions with organic ligands, have received much attention because of their diverse topologies and unique properties in areas such as storage, exchange and conversion[19–21]. Polymerization reactions could also be induced in PCPs, imposing specific nanoconfinement effects on the reaction kinetics and selectivity[22,23]. In our effort to develop an effective methodology for controlling copolymerization, we performed radical polymerization of multiple vinyl monomers accommodated in PCP nanochannels, revealing the compositional changes of monomer units compared with those

obtained from conventional free-radical conditions[24,25]. However, sequence regulation in the resultant copolymers was unsuccessful because of inadequate spatial arrangement of monomers in the nanochannels. Here, we disclose a methodology that utilizes the periodic structure of PCPs for sequence control in radical copolymerization. The structural periodicity of PCPs is precisely transcribed into products at the molecular level, which is distinguished from other examples using PCPs as reaction templates[26–28]. Owing to the high designability of PCPs as well as many applicable monomers, this method can be utilized to fabricate a wide variety of vinyl copolymers with controlled sequences.

## Results

**Preparation of template PCP**. The key to success in this system is pre-immobilization of the vinyl moieties at regular intervals along the one-dimensional nanochannels of a PCP framework via covalent bonding. After the introduction of another free guest monomer into the nanochannels, copolymerization between the immobilized and free monomers provided a copolymer containing a highly periodic sequence because of the regular positioning of monomers in the nanochannels (Fig. 1). To arrange vinyl groups periodically along PCP nanochannels, we focused on a kagomé-type PCP, $[Cu(ipa)]_n$ (**1**; ipa = 5-substituted isophthalates), with one-dimensional nanochannel structure[29–31]. Use of different ipa ligands can allow for the regular alignment of substituents on the ligands along the channels. Here, we employed styrene-3,5-dicarboxylic acid (**S**) as the organic linker to produce [Cu(styrene-3,5-dicarboxylate)]$_n$ (**1S**) for the periodic positioning of the styryl groups in the framework. Mixing of the **S** ligand and $Cu^{2+}$ ion in methanol at room temperature gave the functional host PCP containing coordinated water (**1S⊃H$_2$O**). Single-crystal X-ray diffraction analysis showed that the as-synthesized host has two types of one-dimensional channels with pore diameters of ca 6 Å (hexagonal channels) and 4 Å (triangular channels) along the *c*-axis (Fig. 2a, Supplementary Figs. 1, 2 and Supplementary Table 1). In the larger hexagonal channels, two crystallographically independent styryl groups are arranged alternatively along the channels while the styryl groups of adjacent wall planes point in opposite directions. Thus, styryl groups on each plane of the hexagonal channels were regularly aligned at a distance of 6.8 Å along the channel direction (Fig. 2b), resulting in the periodic arrangement of polymerizable units along the one-dimensional channels.

**Radical copolymerization of acrylonitrile in PCP**. For the copolymerization in **1S**, we used acrylonitrile (**A**) as a guest monomer with molecular dimensions (4.3 × 5.6 Å²) comparable to the size of the hexagonal channel in **1S⊃H$_2$O**. According to the well-established relative reactivities in vinyl copolymerization, the terminal radical group of **A** would preferentially react

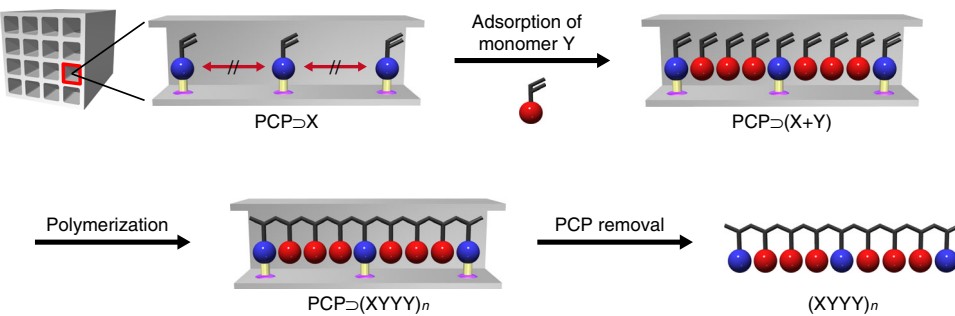

**Fig. 1** Schematic illustration of sequence-regulated radical copolymerization using PCPs. A vinyl monomer (X) is pre-immobilized at a regular interval along the nanochannels of a PCP (PCP⊃X). Another vinyl monomer (Y) is then incorporated into the PCP, giving a host–guest composite (PCP⊃(X + Y)). Polymerization of the monomers in the composite followed by the removal of host structure produces sequence-regulated copolymers reflecting on the periodicity of the PCP nanochannels

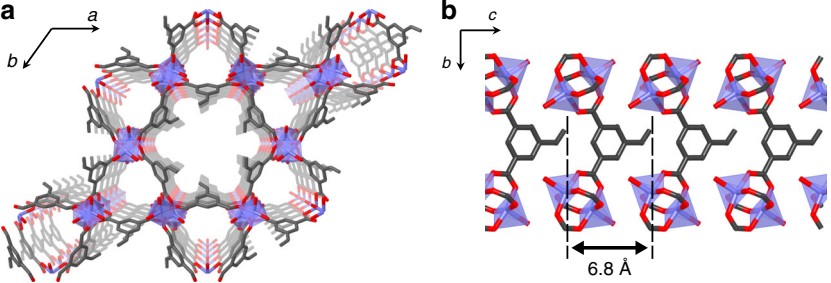

**Fig. 2** Crystal structure of host PCP. Views of the nanochannel structure of **1S**⊃H$_2$O along the *c*-axis (**a**) and *a*-axis (**b**). Atoms: Cu (purple), O (red), C (grey). Styryl groups are periodically aligned along the *c*-axis at a distance of 6.8 Å on each face of the hexagonal channel. H atoms and one of the disordered vinyl moieties have been omitted for clarity

with the styrene derivatives[32], facilitating smooth chain growth along the hexagonal channels of **1S**. As the first step for the copolymerization, the **A** monomer was fully introduced into a degassed host (**1S**) by soaking **1S** in neat **A** and subsequent removal of excess **A** outside the host crystals by evacuation, giving a host–guest composite (**1S**⊃**A**). X-ray powder diffraction (XRPD) measurement showed that the peak positions of **1S**⊃**A** are the same as those of the hydrated host (**1S**⊃H$_2$O), showing the maintenance of the crystalline framework structure upon monomer inclusion (Fig. 3, Supplementary Fig. 3). The **A**/**S** feed ratio in **1S**⊃**A** was found to be 42/58, as determined by $^1$H nuclear magnetic resonance (NMR) measurement (Supplementary Fig. 4). Radical copolymerization in the pores of **1S** was induced by heating **1S**⊃**A** with 2,2′-azobis(isobutyronitrile) (AIBN) at 70 °C for 48 h under nitrogen atmosphere. The lack of change in the XRPD profiles of the composite before and after heating clearly indicated that the crystal structure of the host was retained during the copolymerization (Fig. 3). In addition, scanning electron microscopy and particle size analysis showed the maintenance of morphology and particle size distribution of **1S**, indicating that the copolymerization proceeded only inside the host (Supplementary Figs. 5 and 6). The addition of dilute HCl to the heated composite led to the decomposition of the framework of **1S**, followed by washing with ethanol to give a poly(**A**-*co*-**S**) copolymer denoted as **P1**. In this polymerization system, gram scale synthesis of **P1** was possible if we employed a larger amount of **1S**. In addition, the host PCP framework could be recycled efficiently by the recrystallization of unreacted **S** monomer with Cu ion after the decomposition process (Supplementary Methods).

**Characterizations of copolymer synthesized in PCP.** Formation of the copolymer between **A** and **S** in the channels of **1S** was fully confirmed by infrared, $^1$H NMR and $^{13}$C NMR spectroscopies of **P1** (Supplementary Figs. 7–10). The homopolymer of **A** (poly-acrylonitrile) was not obtained, despite the possible accommodation of **A** in the small triangular pores of **1S**, because of the unpolymerizable monomer arrangement as well as the inaccessibility of the initiator (AIBN) to the narrow channels. In fact, a characteristic peak for polyacrylonitrile was not detectable at $2\theta =$ 17° in the XRPD pattern of **P1** (Supplementary Fig. 11)[33]. Gel permeation chromatography measurement of **P1** after methylation of the carboxylic group[34] of **S** showed that the number-average molecular weight ($M_n$) and polydispersity ($M_w/M_n$) of **P1** were 20,000 and 1.6, respectively (Supplementary Figs. 12 and 13). It should be noted that the compositional ratio in **P1** was calculated to be **A**/**S** = 75/25, as determined by peak integration of $^1$H NMR and elemental analysis (Supplementary Fig. 8), which was greatly different from the initial **A**/**S** feed ratio (42/58) in **1S**⊃**A** (Supplementary Fig. 4). This result contrasted highly with those obtained from copolymerization between free **A** and **S**

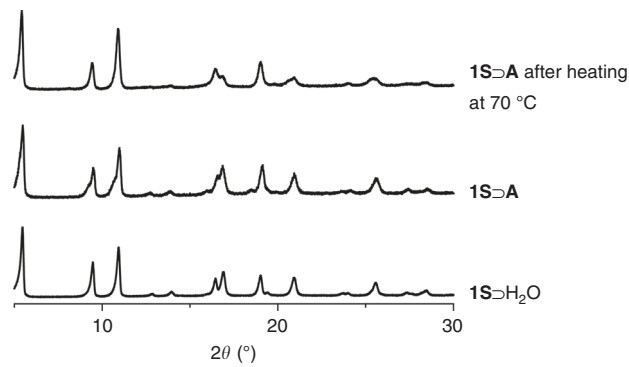

**Fig. 3** XRPD patterns of **1S** with guests. XRPD patterns of **1S**⊃H$_2$O, **1S**⊃**A** and **1S**⊃**A** after heating at 70 °C. The same diffraction patterns were observed in all the samples, showing that the crystal structure of the host was maintained during the monomer adsorption and the copolymerization processes

monomers in *N,N*-dimethylformamide (DMF) solution, where statistically random copolymers (**R1–6**) with **A**/**S** compositions close to their monomer feed ratios were obtained.

**Sequence analysis of copolymer.** Analysis of the $^1$H NMR spectrum in the aromatic region of **P1** implied the feasibility of the sequence regulation in the host–guest copolymerization between **1S** and **A** (Fig. 4). In the spectra of the random copolymers with the composition of **S** unit higher than 17 mol% (**R3–6**), the peak for aromatic protons of **S** was found to broaden with a shift towards higher magnetic field, as was also seen in the spectrum of the homopolymer of **S**, because of the shielding effect from the neighbouring bulky **S** units in the polymer chains[35]. In contrast, the aromatic proton peaks for the **S** units in **P1** were relatively sharp and only detectable at a lower magnetic field ($\delta$ = 7.8–8.4 ppm) despite the adequate content of the **S** unit (25 mol%), suggesting the distribution of solitary **S** units in continuous **A** linkages. $^{13}$C NMR spectroscopy is also a powerful method to prove the regulation of monomer sequences in copolymers[14,36,37]. The predominant triad sequences of **P1** could be determined by analyzing the shape and chemical shift of signals in comparison with those of homopolymers and random copolymers (**R1–6**). Figure 5a shows the spectra in the region for the C=O group of the **S** unit ($\delta$ = 165–168 ppm). In this analysis, a peak for C$_{C=O}$ in the homopolymer of **S** comprising only the **SSS** triad appeared at 166.0 ppm, and this carbonyl peak shifted to lower magnetic field with increasing content of **A** unit in the random copolymers. Note that the C$_{C=O}$ peak for **P1** was found at 166.5 ppm, and the location of this signal was almost the same as that of **R1** (**A**/**S** = 95/5) and **R2** (**A**/**S** = 87/13) with the statistically preferred triad of **ASA**. Thus, the formation of **ASA** should be predominant in the

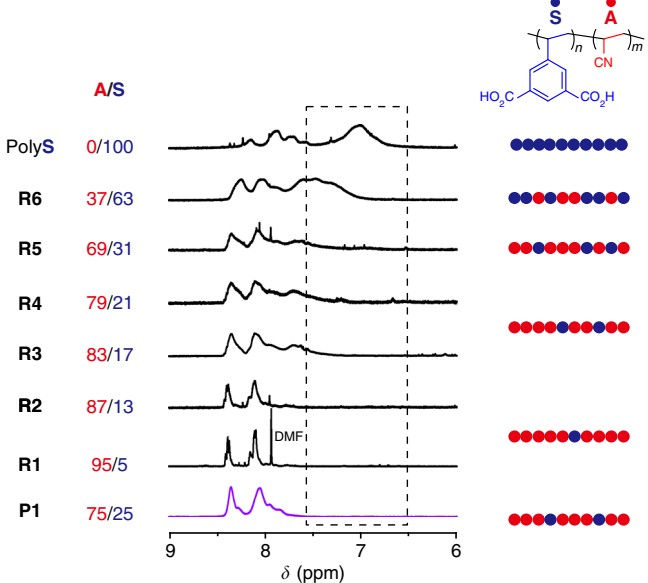

**Fig. 4** Structural characterization of polymers using $^1$H NMR. The $^1$H NMR spectra of (co)polymers synthesized in **1S** (**P1**) and in DMF (**R1-6** and poly**S**). With decreasing **S** unit in polymers prepared in DMF, peaks for the aromatic protons of the **S** unit (6.5–8.5 ppm) shifted to lower magnetic field because of the lower shielding effect around **S** in **A**-rich copolymers. Although the composition of **S** in **P1** is larger than those of **R3** and **R4**, peaks at 6.5–7.6 ppm were not observed, indicating that **S** units in **P1** are solitary among the repeating **A** units

structure of **P1**, as is also supported by the $^1$H NMR result (Fig. 4). In a similar manner, the **A**-centred triad of **P1** could be estimated by analysis of the $C_{CH}$ signals of **A** appearing in the aliphatic region (Fig. 5b). Although copolymers with a lower composition of **A** (**R5**, **R6**) showed featureless broad peaks at 26.5–27.0 ppm, copolymers with high **A** content (**R1**–**R4**) presented three clear peaks in this region. These characteristic peaks were also observed in the spectrum of **P1**, showing that the composition of **P1** is similar to **R1**–**R4** with the predominant triads of **AAA** and **SAA**.

The preferential formation of **ASA**, **AAA** and **SAA** triads in the **P1** chains was strongly supported by the crystal structure of **1S⊃H$_2$O**. As shown in Fig. 2, styryl groups in **1S** are regularly aligned at a distance of 6.8 Å towards the channel direction, which cannot induce the direct reaction between the styryl units. Thus, for the **S**-centred triad, formation of the **ASA** triad was only allowed during the copolymerization in **1S⊃A**. In the same way, the distance between the styryl units along the channels is too long to form the **SAS** triad, resulting in the selective formation of **AAA** and **SAA** for the **A**-centred triad in **P1**. Based on the **A/S** composition (75/25) and detected triads (Fig. 5), the formation of repetitive **SAAA** is the most likely sequence in the structure of **P1**.

Radical copolymerization at different monomer feed ratios provides an insight into the monomer sequence in the resulting copolymer. Loading amount of **A** was controlled by co-introduction with acetonitrile in the host, followed by the copolymerization of **A** with the **S** units in the pores. This polymerization reaction proceeded smoothly, showing that the non-polymerizable solvent molecules do not interfere with the diffusion of the free **A** monomers in the nanochannels. Note that both $^1$H and $^{13}$C NMR spectra of the resulting copolymers were similar to those of **P1**, showing the predominant **ASA**, **AAS** and **AAA** triads in the copolymer chains (Supplementary Figs. 14 and 15). In contrast to the solution copolymerization system, the **A/S** ratio in the copolymers from **1S** was close to that of **P1** (**A/S** = 3/1) regardless of the initial monomer feed ratio, as determined by peak integration

of the $^1$H NMR spectra (Fig. 6 and Supplementary Fig. 14). Independency of the copolymer composition is a key behaviour to prove the regulation of monomer sequence in polymers[11,12], and the constant **A/S** ratio of the copolymers strongly supports the predominant repetitive **SAAA** sequence in the copolymer chains.

**Computational simulation**. To clarify the mechanism of this highly regulated reaction system, we performed a computational analysis on the copolymerization of **A** with **1S** using replica-exchange molecular dynamics (REMD) simulations[38,39] and density functional theory (DFT) calculations[40]. In spite of the possible multiple connectivity between **S** units via **A** chains, REMD simulations suggested that a propagation reaction involving two adjacent styryl groups at different planes of the hexagonal channels causes large steric hindrance for **A** around the terminal **S** radical, resulting in no further polymerization, as is expected from the host crystal structure (Supplementary Figs. 16 and 17a). In contrast, copolymerization of **A** with **1S** can proceed smoothly along the arrays of the styryl units at one face of the channel, picking the regularly positioned styryl moieties into the **P1** chains (Supplementary Fig. 17b and c). Based on this reaction position, we executed DFT calculations on the reaction barriers for the propagation reaction of the terminal **A** radicals in ~**SA**$_n^\bullet$ ($n = 2, 3$; radicals are denoted by a dot) chains with free **A** or styryl monomer immobilized in the host. Taking the difference in the reaction barriers together with the collision frequency estimated from REMD simulations into account, the ~**SAA**$^\bullet$ is likely to react with free **A** monomer to form ~**SAAA**$^\bullet$ radical with very high selectivity (>99%). Note that the resulting ~**SAAA**$^\bullet$ would favour a reaction with the immobilized **S** monomer rather than free **A**, giving a ~**SAAAS**$^\bullet$ chain (Supplementary Fig. 18). These results are fairly consistent with those of the NMR analysis and copolymer composition of **P1**, which shows the validity of the formation of the **SAAA** repeat unit as a predominant sequence.

**Radical copolymerization of another guest monomer in PCP**. The generality of our methodology for the preparation of XYYY copolymers was confirmed using methyl vinyl ketone (**M**) as another guest monomer, where this monomer also has a tendency to crosspropagate with styryl monomers during the copolymerization. Radical polymerization of **M** was performed in **1S**, giving a copolymer (**P2**) with the composition of **M/S** = 76/24, as determined by $^1$H NMR spectroscopy (Supplementary Figs. 19–23). Clearly, this **M/S** ratio is comparable to the **A/S** composition in **P1**. $^1$H and $^{13}$C NMR studies on **P2** showed that the **S** units in **P2** were solitary among **M** units with the probable triad sequences of **MSM**, **MMS** and **MMM**, similar to the case of **P1**. These results strongly suggested that the formation of **P2** in the channels of **1S** would also be regulated to give a repetitive **SMMM** sequence (Supplementary Figs. 24 and 25).

**Discussion**
We have established an approach to sequence-controlled radical copolymerization using the regular structure of a PCP as a template. The capability of PCPs to transcribe their periodic structure into a product was demonstrated. The **S** monomer anchored into the host PCP was copolymerized with guest vinyl monomers, producing copolymers with a predominant XYYY sequence. Our methodology exploiting crystalline templates constructed by self-assembly of metal ions and ligands provides an easy access to the sequence-regulated polymerization with scalability and recyclability. In addition, this work achieved successfully sequence-controlled radical copolymerization using an artificial polymeric template. Preparation of polymeric templates with well-defined structures is generally quite difficult because of the requirement for

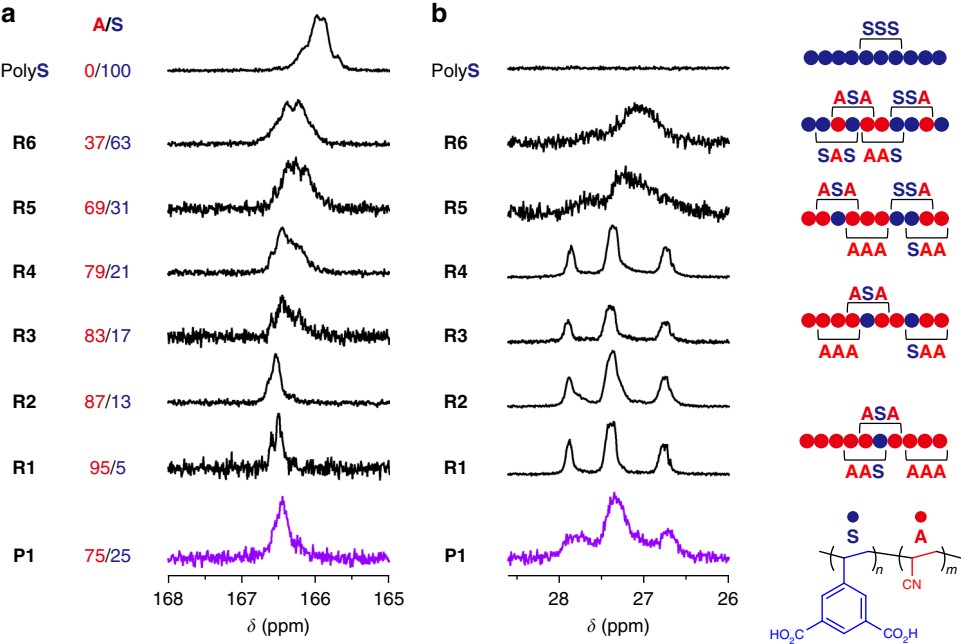

**Fig. 5** Structural characterization of polymers using $^{13}C$ NMR. $^{13}C$ NMR spectra focusing on carbonyl (**a**) and methine carbons (**b**) of polymers, from which **S**- and **A**-centred triads can be analyzed, respectively. Plausible monomer sequences of random copolymers (**R1**–**6**) are shown together with their NMR spectra. The predominant triads of **P1** could be estimated to be **ASA**, **AAS** and **AAA** by comparison of the peak shapes and peak positions for **P1**, random copolymers (**R1**–**6**) and poly**S**

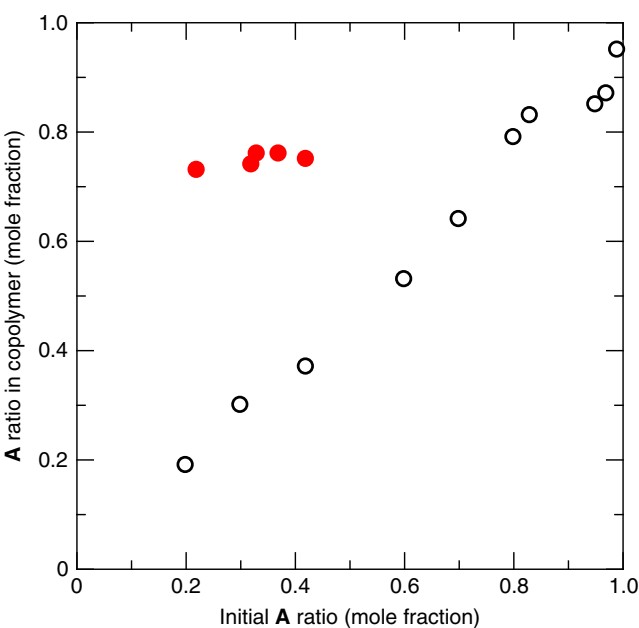

**Fig. 6** Copolymer composition plots for the copolymerization of **A** with **S**. A series of **A** ratio in the generated copolymers were plotted against their initial **A** ratio for the copolymerization of **A** with **S** in **1S** (filled red circle) and in DMF (open circle). Note that the maximum initial ratio of **A** is 0.42 in the channels of **1S** due to the limitation of introduction amount of **A** into the host

complicated organic synthesis with many reaction steps. Moreover, use of small templates in solution copolymerizations often led to undesired crosslinking reactions[41]. Our methodology using PCP templates could overcome these problems, showing the remarkable advancement in this field. Owing to the high designability of PCPs, predetermination of monomer units in PCP frameworks can allow for materials platforms to accomplish many other periodic sequences for copolymers. We believe that this study opens the door to the utilization of crystalline templates for controlled monomer sequences in copolymers.

## Methods

**Synthesis of 1S⊃H$_2$O**. Single crystals of **1S⊃H$_2$O** were synthesized as follows. Slow diffusion of a methanol solution of **S** into a methanol solution of Cu(NO$_3$)$_2$·3H$_2$O and pyridine afforded sky-blue polycrystals in a few days. They were cut into small single crystals and then used for the single-crystal X-ray diffraction. A crystalline powder sample of **1S⊃H$_2$O** was prepared as follows. Methanol (200 mL) solutions of **S** (4.00 g, 20.8 mmol) and Cu(NO$_3$)$_2$·3H$_2$O (5.03 g, 20.8 mmol) were mixed and stirred for 12 h after the addition of pyridine (1.68 mL, 20.8 mmol). The mother liquor was decanted and the light blue solid was washed with methanol to remove unreacted materials. Light blue powder (2.85 g) was obtained after drying under vacuum for 2 h at room temperature.

**Copolymerization of guest vinyl monomers with S in 1S**. A typical procedure for copolymerization of **A** and **S** in the nanochannels of **1S**. The degassed host compound **1S** was obtained (168 mg) by evacuation (<0.2 kPa) at 110 °C for 2 h, and was then immersed in the solution of **A** (1 mL) and AIBN (3 mg) in a 30-mL flask under nitrogen atmosphere. The mixture was left for 30 min to incorporate the monomer and the initiator into **1S**. The host–guest composite (**1S⊃A**) was obtained after removing excess monomer outside the PCP crystals completely by evacuation at 8.0 kPa. **1S⊃A** was heated at 70 °C for 48 h to copolymerize **A** with the styryl groups immobilized in the frameworks of **1S**. The reaction was quenched by the addition of methanol. After removing the supernatant solution and drying the compound, 1 M HCl was added to decompose the PCP framework of **1S** as well as to acidify the carboxylate group of the resulting copolymer. **P1** was obtained by washing the product with ethanol, followed by drying under reduced pressure at room temperature (35 mg). The C/N atomic ratio of **P1** was 5.194/0.8307 (C/N mass ratio: 62.33/11.63), as calculated from elemental analysis, showing that the **A**/**S** ratio in **P1** was determined to be 75:25. Gram scale synthesis of **P1** was conducted using 5.40 g of **1S**, generating 1.09 g of **P1**. Polymerization of **A** in **1S** with different loading amounts of monomer was also performed according to the procedure described above after the co-introduction of acetonitrile into the host. For the synthesis of **P2** in **1S**, we used the same procedure other than the polymerization temperature (100 °C) and washing solvent (ethyl acetate).

**Data availability**. The X-ray crystallographic coordinates for the structure reported in this article have been deposited at the Cambridge Crystallographic Data Centre (CCDC), under deposition number CCDC 1430041. These data can be obtained free of charge from the Cambridge Crystallographic Data Centre via www.ccdc.cam.ac.uk/data_request/cif. All relevant data are available from the authors upon reasonable requests.

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

## Acknowledgements

This work was supported by CREST-JST (JPMJCR1321) and JSPS KAKENHI Grant Number JP16H06517 (Coordination Asymmetry). The authors would like to thank Dr. Hiroshi Sato (The University of Tokyo) for valuable technical discussions. We also acknowledge Dr. Nobuhiro Yasuda (Japan Synchrotron Radiation Research Institute, SPring-8) for single-crystal X-ray diffraction measurements.

## Author contributions

T.U. conceived and directed the project. T.U. and S.M. designed and performed the experiments. N.O. contributed to single-crystal X-ray diffraction analysis. M.T. and M.N. performed the computational analysis. S.M., S.K. and T.U. discussed the results and wrote the paper.

## Additional information

**Competing interests:** The authors declare no competing financial interests.

