## [Peer Review File · Nature Communications]

Reviewers' comments:

Reviewer #1 (Remarks to the Author):

This manuscript reports the synthesis of periodic copolymers by radical copolymerization in a confined environment. In brief, a styrenic derivative containing two carboxylic acid groups was periodically immobilized in the nanochannels of a porous coordination polymer. Afterwards, this immobilized monomer was copolymerized with a host comonomer, i.e. acrylonitrile. This is an interesting concept that aims to highlight the relevance of confinement in sequence-regulated copolymerizations. That said, I do not think that the experimental data reported in this (relatively concise) manuscript justify publication in a multidisciplinary journal such as Nature Communications. First of all, the analytical characterization of the formed copolymers is quite thin. I agree with the authors that Figure 4 tends to indicate a periodic sequence. However, there is obviously still a relatively high degree of randomness in these copolymers. In my opinion, the formed copolymers would deserve a deeper analysis using other analytical methods than NMR (e.g. advanced chromatography techniques or mass spec). More importantly, the study is too short and would benefit from additional experiments (e.g. copolymerization with other host monomers, other polymerization conditions, investigation of co-solvent effects). Furthermore, a periodic copolymer by itself is not a huge progress beyond the state-of-the-art. It would be probably more impressive if the authors could immobilize more than one monomer in their network. For instance, the synthesis of diperiodic or triperiodic copolymers would be a significant achievement. In its current form, I believe that present manuscript can be published in a good specialized journal if the periodicity is confirmed by other analytical methods.

Reviewer #2 (Remarks to the Author):

This manuscript from Uemura and Kitagawa describes the copolymerization of a styrene derivative (S) and acrylonitrile (A) in the channels of a crystalline porous coordination polymer. The styrene derivative, styrene-3,5-dicarboxylic acid, is employed as a ligand in the PCP, which leads to well-defined spacing of the olefinic monomer units. The authors find that polymerization produces copolymers that have no detectable consecutive S units, and based on the monomer ratio of 75:25 A:S they propose a well-defined sequence of SAAA. This approach offers a unique way to template the synthesis of sequence controlled polymers, and, in my opinion, should be published in Nature Communications following minor revisions.

- 1) The authors should comment on defects in their PCP, and how defects would impact the polymerization. For example, depending on the size of the crystals use, the surfaces could provide sites for less controlled polymerization.
- 2) It would be nice to see a few experiments conducts at different A loadings. In principle, the same polymers should be produced regardless of the amount of A loading into the PCP.
- 3) Here the authors used 42:58 A:S. What happened to the remaining S after the reaction? Did it not react at all?
- 4) The authors should comment on the average length and length distribution of the channels, and describe how these parameters might translate to polymer dispersity.
- 5) Could images of the crystals before and after loading and polymerization be provided in the SI?
- 6) The authors should add a discussion regarding the scalability and yield of this approach. How much polymer can reasonably be produced using this method? Can the PCP be re-generated after breakdown and removal of polymer by adding in more S?
- 7) It would be interesting and helpful to polymerize a longer monomer and demonstrate that fewer monomers are incorporated. Butadiene could be a candidate.

Reviewer #3 (Remarks to the Author):

Synthesis of sequence controlled polymers is in fashion these days. As such, the presented approach is interesting as it considers the synthesis of large amounts of polymer via a clever templating approach. However, the manuscript is missing the required detailed analysis and characterization to clearly proof and support all claims made.

* Characterization has mainly been conducted by NMR. In order to take reasonable information of out it, it would be good to compare the NMR spectra to those obtained by classical copolymerization.

* As for the success of the polymerization in 1S \rightarrow A, have the authors considered IR analysis?

* In order to clearly prove the sequences, which is the central part of the paper, mass-spectrometric analysis would be required!

* Can "chain stoppers" be introduced by building a PCP of a mixture of styrene-3,5-dicarboxylic acid and 3,5-dicarboxylic acid?

* How about a better compatible monomer pair? Choosing monomers that withstand homopolymerization would be an advancing step in the concept.

* Can the authors document the preferential formation of ASA, AAA and SAA triads by ^{13}C NMR analysis?

Considering these comments, I recommend to reject the manuscript in its current form, as it does not justify the careful and exhaustive characterization required to support publication in Nature Communications.

Response to Reviewers' comments:

Reviewer #1 (Remarks to the Author):

This manuscript reports the synthesis of periodic copolymers by radical copolymerization in a confined environment. In brief, a styrenic derivative containing two carboxylic acid groups was periodically immobilized in the nanochannels of a porous coordination polymer. Afterwards, this immobilized monomer was copolymerized with a host comonomer, i.e. acrylonitrile. This is an interesting concept that aims to highlight the relevance of confinement in sequence-regulated copolymerizations. That said, I do not think that the experimental data reported in this (relatively concise) manuscript justify publication in a multidisciplinary journal such as Nature Communications.

1) First of all, the analytical characterization of the formed copolymers is quite thin. I agree with the authors that Figure 4 tends to indicate a periodic sequence. However, there is obviously still a relatively high degree of randomness in these copolymers. In my opinion, the formed copolymers would deserve a deeper analysis using other analytical methods than NMR (e.g. advanced chromatography techniques or mass spec).

As the reviewer pointed out, advanced chromatography and mass spectroscopy techniques are often useful for analyzing the sequence of copolymers. Unfortunately, these analytical methods are not applicable to the analysis of the copolymer (**P1**) obtained in this work, because of several inherent problems, while NMR and computational analyses clearly indicated the controlled repetitive sequence in **P1**. For example, acrylonitrile (**A**) units in **P1** are carbonized by heating, so that pyrolysis chromatography cannot be employed. Low solubility of **P1** in common organic solvents impedes the utility of MALDI-TOF-MS that is often used for the sequence analysis of copolymers. For the MALDI-TOF-MS analysis, polymer samples should be co-crystallized with matrix using volatile solvents (S. D. Hanton *et al.* In *MALDI Mass Spectrometry for Synthetic Polymer Analysis*; L. Li, Ed.; Wiley: Hoboken NJ, 2010; pp 267–288.). However, **P1** was insoluble in any common volatile organic solvents, such as THF, MeOH, and CHCl₃, because of the high content of polar **A** and COOH group in the structure. This polymer can be dissolved in DMSO and was thus analyzed for MALDI-TOF-MS using the DMSO solution. However, we could obtain only low resolution spectra, which did not allow for further analysis of the repeated sequence of the copolymer. Our attempt using solvent-free **P1** sample for MALDI-TOF-MS did not work as well.

In addition to the low solubility of **P1**, the high molecular weight of this polymer prevented the MALDI-TOF-MS analysis because of lower resolution in the higher molecular weight region. Usually, sequence analysis of copolymers using MALDI-TOF-MS is performed for oligomeric copolymers with molecular weight of several thousands, so that copolymers with high molecular weight ($M_w > 10,000$) are not well-analyzed (Y. Hibi, *et al.*, *Angew. Chem. Int. Ed.*, **2011**, *50*, 7434 – 7437. K. Satoh *et al.*, *J. Am. Chem. Soc.*, **2010**, *132*, 10003–10005.). Molecular weight of **P1** was found to be 20,000 with less oligomeric fraction (the relatively narrow polydispersity: $M_w/M_n = 1.6$), which hinders the MS spectroscopy.

Another important factor for the MALDI-TOF-MS measurement is the requirement of well-defined terminal structures of copolymers. Note that, in our copolymerization system, use of AIBN as a radical initiator promoted free-radical polymerization, which resulted in uncontrolled random terminal structures of **P1** chains. Actually, the number of **A** units at initial and end terminals cannot be regulated in the polymer chains. Due to the limitation of MALDI-TOF-MS, polymers with only defined terminal and repetitive units can be analyzed, which is usually achieved by living radical polymerization systems (K. Satoh *et al.*, *J. Am. Chem. Soc.*, **2010**, *132*, 10003–10005. K. Nishimori *et al.*, *Macromol. Rapid. Commun.*, **2016**, *37*, 1414-1420.).

For these reasons, we alternatively analyzed the composition and monomer sequence of **P1** using NMR spectroscopy, elemental analysis, MD simulations, and DFT calculations. In addition to these analyses, as the reviewer suggested, we obtained new data about the constant monomer composition (**A:S** = 3:1) in **P1** independent of the monomer feed ratio, which strongly indicates the regulation of monomer sequence in the generated copolymers. As we describe later in this letter, the periodic monomer sequences of copolymers are often supported by the relationship between monomer feed ratio and copolymer composition. (K. Satoh *et al.*, *J. Am. Chem. Soc.*, **2010**, *132*, 10003–10005. S. Banerjee *et al.*, *ACS Macro Lett.*, **2016**, *5*, 1232–1236.)

2) More importantly, the study is too short and would benefit from additional experiments (e.g. copolymerization with other host monomers, other polymerization conditions, investigation of co-solvent effects).

We appreciate the valuable comments on additional experiments from this reviewer. Results of the new experiments have provided the deeper understanding for our polymerization system, which powerfully supports the formation of periodic **SAAA** sequence in the resulting copolymer.

As described above, we performed copolymerization in **1S** under different monomer feed ratio. It was found that the monomer composition of the generated copolymers were close to that of **P1** (**A/S** = 3/1), even when the initial **A** ratio was small (**A/S** = 22/78, initial feed mole ratio). This polymerization behavior with the monomer compositions independent of the feed ratio is an important key to assure the generation of copolymers with periodic monomer sequences (K. Satoh *et al.*, *J. Am. Chem. Soc.*, **2010**, *132*, 10003–10005. S. Banerjee *et al.*, *ACS Macro Lett.*, **2016**, *5*, 1232–1236.). In addition, study on the triad structures of these new copolymers using ¹³C NMR spectroscopy showed that **SAAA** repetitive sequence is predominantly formed, strongly supports the mechanism described in the manuscript.

Investigation of co-solvent effects on polymerization offered new insight into the monomer dispersion and the reactivity of monomer in the nanochannels. Here we employed acetonitrile as a non-polymerizable co-solvent because of the structural similarity to **A** as well as the tunability of monomer loading amount in the channels. Copolymerization of **A** with styryl units in the pore successfully proceeded even with the existence of acetonitrile, showing that diffusion of **A** was not suppressed by the co-existence of other solvent molecules.

We have added these new data in the revised manuscript to understand and support the mechanism of our polymerization (page 11, lines 5-18, Supplementary Figs. 14 and 15).

3) Furthermore, a periodic copolymer by itself is not a huge progress beyond the state-of-the-art. It would be probably more impressive if the authors could immobilize more than one monomer in their network. For instance, the synthesis of diperiodic or triperiodic copolymers would be a significant achievement.

As the reviewer suggested, further investigation on the copolymerization using other hosts would provide the importance and applicability of this copolymerization system. We believe that our work reported in this manuscript will open up a new methodology to demonstrate that the periodicity of PCP can be transferred to the product. Thanks to the structural diversity of PCP frameworks, we are potentially able to prepare various template scaffolds for copolymerization with different periodic distances of immobilized monomers, multiple periodicities, other immobilized monomer species, and so on.

We would like to emphasize that the methodology reported here showed considerable progress from the viewpoint of template polymerization system. This work is the first example that could achieve sequence-controlled radical copolymerization using artificial polymeric templates. Regulation of monomer sequences via template radical polymerization has not been accomplished yet, except for the case using small template molecules with specific monomer pairs (Y. Hibi *et al.*, *Angew. Chem. Int. Ed.*, **2011**, *50*, 7434. Y. Hibi *et al.*, *Polym. Chem.*, **2011**, *1*, 341. M. Ouchi *et al.*, *Angew. Chem. Int. Ed.*, **2016**, *55*, 14584.). To begin with, preparation of polymeric templates with well-defined structures is quite difficult because of the requirement for complicated organic synthesis with many reaction steps (Y. Hibi *et al.*, *Angew. Chem. Int. Ed.*, **2011**, *50*, 7434. J. Niu *et al.*, *Nat. Chem.*, **2013**, *5*, 282–292.) Moreover, template copolymerization in solution system often faces undesired crosslinking reactions (R. Jantas, *Acta Polym.*, **1991**, *42*, 539-544.). However, our methodology would overcome these problems by exploiting a self-assembled crystalline coordination compound with nanochannels as a template material, which enabled us to construct polymeric template with complete periodicity as well as inhibit the crosslinking reaction by accommodating the propagating radicals in the nanochannels. These features are clearly different from conventional template polymerization systems, showing the remarkable advancement in this field.

Reviewer #2 (Remarks to the Author):

This manuscript from Uemura and Kitagawa describes the copolymerization of a styrene derivative (S) and acrylonitrile (A) in the channels of a crystalline porous coordination polymer. The styrene derivative, styrene-3,5-dicarboxylic acid, is employed as a ligand in the PCP, which leads to well-defined spacing of the olefinic monomer units. The authors find that polymerization produces copolymers that have no detectable consecutive S units, and based on the monomer ratio of 75:25 A:S they propose a well-defined sequence of SAAA. This approach offers a unique way to template the synthesis of sequence controlled polymers, and, in my opinion, should be published in Nature Communications following minor revisions.

1) The authors should comment on defects in their PCP, and how defects would impact the polymerization. For example, depending on the size of the crystals used, the surfaces could provide sites for less controlled polymerization.

Crystallinity of PCPs would affect the polymerization behavior if PCPs are used as hosts for radical polymerization. In general, use of the PCPs with lower crystallinity generates polymers with lower yield and molecular weight due to incidental quenching of polymerization at defect sites, especially at Cu corners, with redox activity. In the current polymerization system reported in our manuscript, the effect of defect sites in the PCP is almost ignorable, judging from the high crystallinity observed in XRPD measurements, which eventually leads to the generation of copolymer (**P1**) with high molecular weight and good yield.

Additionally, we do not expect any effect of crystal surface on the copolymerization in this work because the copolymerization reaction proceeded only inside the pore of the host PCP. The size of PCP crystals used was much larger than the resulting polymer chain length estimated from the molecular weight.

2) It would be nice to see a few experiments conducted at different A loadings. In principle, the same polymers should be produced regardless of the amount of A loading into the PCP.

As we answered for the comment from the reviewer 1, we have performed a new experiment using different loading amount of **A** monomer. We found that radical copolymerization at even different initial monomer molar ratios could produce copolymers with the constant monomer ratio of **A** to **S** (3 to 1) regardless of the initial monomer ratio. In addition, the structure of the copolymers was the same to that of **P1**, as determined by the ^1H and ^{13}C NMR spectroscopies. We appreciate the valuable comment from this reviewer, and included these explanations in the revised version of the manuscript (page 11, lines 5-18, Supplementary Figs. 14 and 15), strongly supporting the controlled sequence in **P1**.

3) Here the authors used 42:58 A:S. What happened to the remaining S after the reaction? Did it not react at all?

As we described in the manuscript, not all the **S** monomers tethered at the host can participate in the copolymerization. Due to steric demand, the copolymerization reaction proceeds along the one-dimensional nanochannel to pick up only the immobilized S monomers at one face of the hexagonal planes, as was demonstrated by MD simulations. Considering the channel size of **1S**, only single polymer chains can be generated in each channel. Thus, one sixth (17%) of **S** units in the channel are at most able to be converted during copolymerization theoretically. In our experiment, 15% of **S** units have reacted with **A** monomers to form a polymer in the channels, which was almost consistent with the theoretical value.

4) The authors should comment on the average length and length distribution of the channels, and describe how these parameters might translate to polymer dispersity.

5) Could images of the crystals before and after loading and polymerization be provided in the SI?

To answer the questions 4 and 5 from this reviewer, we performed particle size distribution analysis and SEM measurement of the PCP crystals before and after the copolymerization reaction. In these analyses, no aggregation of the host crystals was observed after the copolymerization. In addition, morphology of the crystals was retained during the reaction, indicating that the copolymerization reaction proceeded only inside the pores of PCP crystals. Because of the one-dimensional nanochannel structure of **1S**, the average length of the channels in the single particles would be several micrometers, which is much longer than the estimated maximum length of the completely extended copolymer chains (ca. 66 nm, $M_n = 20,000$). There is no correlation between the channel length and the molecular weight of resulting copolymer because the radical initiator (AIBN) was included only inside the nanochannels of host and initiated the polymerization reaction homogeneously throughout the channels in current system. In this revision, we included these explanations (page 7, line 17 – page 8, line 2), and added particle size distribution and SEM images of the host crystals before and after the polymerization into Supplementary Information (Supplementary Figs. 5 and 6).

6) The authors should add a discussion regarding the scalability and yield of this approach. How much polymer can reasonably be produced using this method? Can the PCP be re-generated after breakdown and removal of polymer by adding in more S?

Thank you for the valuable comment. In this approach using PCPs, we can scale up the reaction as long as the host PCP can be prepared. In the manuscript, only a small scale reaction was described; however, we have successfully performed gram-scale reactions using PCPs, producing

sufficient amount of polymers. With regard to the reusability of PCPs after reaction, this PCP could be re-generated using unreacted **S** after purification, followed by complexation with Cu ion.

7) It would be interesting and helpful to polymerize a longer monomer and demonstrate that fewer monomers are incorporated. Butadiene could be a candidate.

As the reviewer suggested, we tried to perform radical copolymerization of a butadiene (2,5-dimethyl-1,3-butadiene: DMB) with styryl monomer in the channels of **1S**. We have already performed the homopolmerization of DMB in PCPs within a PCP (T. Uemura *et al.*, *Chem. Commun.*, **2015**, *51*, 9892-9895.); however no homo- and co-polymer were obtained when using **1S** as a host probably because of the decrease in the reactivity of the monomer in the narrow pores.

Reviewer #3 (Remarks to the Author):

Synthesis of sequence controlled polymers is in fashion these days. As such, the presented approach is interesting as it considers the synthesis of large amounts of polymer via a clever templating approach. However, the manuscript is missing the required detailed analysis and characterization to clearly prove and support all claims made.

1) Characterization has mainly been conducted by NMR. In order to take reasonable information of it, it would be good to compare the NMR spectra to those obtained by classical copolymerization.

As the reviewer suggested, we compared the ^1H and ^{13}C NMR spectra of **P1** to those of random copolymers with similar composition obtained by classical solution polymerization, showing clear differences due to the regulated periodic monomer sequence in **P1**. In the ^1H NMR measurement, broad signals assigned to the three aromatic protons of the **S** unit in **P1** did not appear at higher magnetic field region when compared to those observed in the random copolymer with similar compositions, indicating the distribution of solitary **S** units in continuous **A** linkages (page 9, lines 4-13, Fig. 4). Fig. 5 shows the carbonyl carbon peak of **S** and the methine carbon peak of **A** in ^{13}C NMR spectra of copolymers. The predominant triad sequences of **P1** can be determined by analysing the shape and chemical shift of signals in comparison with those of homopolymers and random copolymers, showing the preferential formation of **ASA**, **AAA** and **SAA** triads in the **P1** chains (page 9, line 13 – page 10, line 12).

2) As for the success of the polymerization in $1\text{S}\supset\text{A}$, have the authors considered IR analysis?

Thank you for your comment. In order to confirm the success of the polymerization and to identify the product, we have measured IR spectroscopy after the heating of $1\text{S}\supset\text{A}$ followed by the liberation of polymeric product from the host PCP. As was shown in Supplementary Fig. 7, characteristic peaks for C=O, C≡N and O–H stretching vibrations were observed at 1713, 2244, and 2345–3700 cm^{-1} , respectively, indicating that **P1** contained both **A** and **S** units in its structure.

3) In order to clearly prove the sequences, which is the central part of the paper, mass-spectrometric analysis would be required!

As we commented to reviewer 1, mass spectrometry was not effective for the sequence analysis of the obtained copolymer in this study because of the several intrinsic problems for this technique (only applicable for oligomers with the molecular weight less than 10,000, limitation for the solvents, and necessity for the regulation of terminal groups). Alternatively, other powerful analyses and experimental results, including NMR spectroscopy, copolymer composition, crystal structure analysis, and computational simulations, strongly indicated the formation of repetitive

SAAA unit in **P1**. In addition, we performed a new experiment for copolymerization in **1S** under different monomer feed ratio. The monomer composition of all the generated copolymers (**A/S** = 3/1) was found to be independent of the feed ratio, which also assure the generation of copolymers with periodic monomer sequences. Moreover, we confirmed that both ^1H and ^{13}C NMR spectra of these copolymers were very similar to those of **P1**, indicating that the copolymers also had **ASA**, **AAS** and **AAA** as predominant triads. We have added these explanation in the revised version (page 11, lines 5-18, Supplementary Figs. 14 and 15).

4) Can "chain stoppers" be introduced by building a PCP of a mixture of styrene-3,5-dicarboxylic acid and 3,5-dicarboxylic acid?

It is an interesting idea to incorporate "chain stoppers" in the framework of PCPs as the reviewer suggested. We have utilized 3,5-benzenedicarboxylic for the preparation of isostructural **1S** and performed polymerization of vinyl monomers, such as styrene, methyl methacrylate and **A** in this framework. Unfortunately, this ligand does not act as a chain stopper and thus the PCP with 3,5-benzenedicarboxyalte could produce many vinyl polymers in the nanochannels. However, it would be promising to use another chain stopper for the selective synthesis of oligomeric products. Introduction of a ligand with bulky substituents would stop propagating reaction effectively at regular interval, forming oligomers with controlled chain length.

5) How about a better compatible monomer pair? Choosing monomers that withstand homopolymerization would be an advancing step in the concept.

In our polymerization system, it is crucial to choose guest monomers that have the preferential reactivity with styrene. Otherwise, propagating chain would skip the reaction with immobilized **S** in **1S**, resulting in the copolymers with random monomer sequence. Thus, in this study, we employed **A** and methyl vinyl ketone (**M**) as comonomers, because both monomers have a tendency to crosspropagate with styrene (E. Ōsawa *et al.*, *Makromol. Chem.* **1965**, *83*, 100–112. S. G. Bond *et al.*, *Polymer*, **1998**, *39*, 6875-6882.). We described this explanation in the manuscript (page 7, lines 3-6; page 13, lines 5-7). As a better compatible monomer pair, we have used maleic anhydride (MA) as a guest monomer, and performed the polymerization of MA in **1S** in this revision. MA is an electron-accepting monomer, and thus gives an alternating copolymer by the copolymerization between MA and styrene derivatives (M. C. Davies *et al.*, *Polymer*, **2005**, *46*, 1739-1753). However, in our experiment, the radical copolymerization did not effectively proceed probably because of the large monomer size.

6) Can the authors document the preferential formation of ASA, AAA and SAA triads by ^{13}C NMR analysis?

We were able to make out the preferential formation of **ASA**, **AAA** and **SAA** triads by ^{13}C NMR analysis based on the comparison of the NMR spectrum of **P1** to those of homopolymers and random copolymers obtained by solution copolymerization. Fig. 5a shows the spectra in the region for the C=O group of the **S** unit ($\delta = 165\text{--}168$ ppm). In this analysis, a peak for $\text{C}_{\text{C=O}}$ in the homopolymer of **S** comprising only the **SSS** triad appeared at 166.0 ppm, and this carbonyl peak shifted to lower magnetic field with increasing content of **A** unit in the random copolymers. Note that the $\text{C}_{\text{C=O}}$ peak for **P1** was found at 166.5 ppm, and the location of this signal was almost the same as those of **R1** (**A/S** = 95/5) and **R2** (**A/S** = 87/13) with the statistically preferred triad of **ASA**. Thus, **ASA** should be the predominant **S**-centered triad in the structure of **P1**, as was also supported by the ^1H NMR result (Fig. 4). In a similar manner, the **A**-centred triad of **P1** could be estimated by analysis of the C_{CH} signals of **A** appearing in the aliphatic region (Fig. 5b). Although copolymers with a lower composition of **A** (**R5**, **R6**) showed featureless broad peaks at 26.5–27.0 ppm, copolymers with high **A** content (**R1–R4**) presented three clear peaks in this region. These characteristic peaks were also observed in the spectrum of **P1**, showing that the composition of **P1** is similar to **R1–R4** with the predominant triads of **AAA** and **SAA**. We can emphasize that the preferential formation of **ASA**, **AAA** and **SAA** triads in the **P1** chains was also strongly supported by the crystal structure and computational simulations. These explanations were found in the main text (page 9, line 13 – page 10, line 12).

Reviewers' comments:

Reviewer #1 (Remarks to the Author):

The authors have made some efforts to strengthen their manuscript. As indicated in my earlier report (and also in those of other reviewers), additional analytical proofs would have been beneficial to support their claims. However, I understand their arguments regarding the applicability of other techniques such as MS and I agree that it is not an easy task in the present case.

I also think that the use of different feed ratio in P1 synthesis is an important experiment to support the conclusions of this paper. I would actually highlight these interesting results in the main text. For instance, Supplementary Figures 14 and 15 shall be transferred to the main text. The authors could draw a composite figure showing the effect of comonomer feed on P1 composition. To make it even more clear for the readers, this figure could be completed by a classical composition plot, i.e. copolymer composition F versus comonomer feed f . However, 3 points are probably not enough for such a graphic. It would be interesting to have at least 5/6 points on it. Last but not least, this plot could be compared for templated and non-templated copolymerizations. That would clearly highlight the influence of the template.

I also understand the arguments of the authors regarding their achievements from the viewpoint of a template polymerization. However, I still believe that it would be interesting for the readers to feel the applicability of this method for the synthesis of more elaborated sequence-controlled polymers. In that regard, I recommend to split the results and discussion part into two distinct sections and to use the discussion section to discuss the pro and cons of the method.

With all these modifications, I believe that this manuscript could become suitable for publication in Nature Communications.

Reviewer #2 (Remarks to the Author):

The authors have done an adequate job of responding to the reviewers' comments. The finding that variation in feed ratio provides polymers with the same monomer composition is compelling evidence that the method works as described.

Prior to publication, however, the authors should provide details of their claims that (a) they produced polymers on the gram scale, and (b) they could recycle the framework. They state in their rebuttal that both of these feats are possible. In my opinion, this manuscript would be greatly improved if they provide proof.

Otherwise, I tend to agree with sentiments of Reviewer #1: small scale synthesis of a periodic copolymer is not by itself very exciting, since there are other methods to make analogous structures, and there is no compelling function demonstrated for these specific polymers. If the authors could show evidence of true scalability and recycling of the framework, it would be a significant advance as other templated methods are greatly limited in terms of scale. Hopefully these results could be added.

Reviewer #3 (Remarks to the Author):

Having looked at the revision, the authors have included substantial new experimental data. However, they still do not address the fundamental limitation of the present paper, which they also state in their comments: "... SAAA repetitive sequence is predominantly formed, ... ", but not exclusively.

As such, I would recommend to submit the manuscript to a more specialized journal.

Response to Reviewers' comments:

Reviewer #1

Comment 1: I also think that the use of different feed ratio in P1 synthesis is an important experiment to support the conclusions of this paper. I would actually highlight these interesting results in the main text. For instance, Supplementary Figures 14 and 15 shall be transferred to the main text. The authors could draw a composite figure showing the effect of comonomer feed on P1 composition. To make it even more clear for the readers, this figure could be completed by a classical composition plot, i.e. copolymer composition F versus comonomer feed f. However, 3 points are probably not enough for such a graphic. It would be interesting to have at least 5/6 points on it. Last but not least, this plot could be compared for templated and non-templated copolymerizations. That would clearly highlight the influence of the template.

Answer 1: We appreciate the valuable comments from this reviewer. In order to emphasize the evidence of controlled sequence in **P1**, additional experiments for the copolymerization in **1S** with different initial monomer ratios were performed (totally 5 points) to draw the copolymer composition plots. We confirmed again that the **A/S** ratios in the copolymer were very similar ($A/S=3/1$), independent of the initial monomer ratio. The obtained data was compared with those of the solution copolymerization system in Fig 6, showing a clear difference between these two methods. We included these explanations and the new Figure for the copolymer composition plots into the main text of the revised manuscript (page 12, lines 1-7, Fig 6), as Figures 14 and 15 were left remained in Supplementary Information.

Comment 2: I also understand the arguments of the authors regarding their achievements from the viewpoint of a template polymerization. However, I still believe that it would be interesting for the readers to feel the applicability of this method for the synthesis of more elaborated sequence-controlled polymers. In that regard, I recommend to split the results and discussion part into two distinct sections and to use the discussion section to discuss the pro and cons of the method.

Answer 2: Thank you for the useful advice. In this revision, we split the results and discussion part into two distinct sections according to the reviewer's suggestion. In the discussion part, we described more clearly the unique features and applicability of the methodology demonstrated in our work (page 14, line 10 – page 15, line 1).

Reviewer #2

Comment 1: Prior to publication, however, the authors should provide details of their claims that (a) they produced polymers on the gram scale, and (b) they could recycle the framework. They state in their rebuttal that both of these feats are possible. In my opinion, this manuscript would be greatly improved if they provide proof.

Otherwise, I tend to agree with sentiments of Reviewer #1: small scale synthesis of a periodic copolymer is not by itself very exciting, since there are other methods to make analogous structures, and there is no compelling function demonstrated for these specific polymers. If the authors could show evidence of true scalability and recycling of the framework, it would be a significant advance as other templated methods are greatly limited in terms of scale. Hopefully these results could be added.

Answer 1: In order to show the evidence of scalability for our methodology, we performed the radical copolymerization of **A** with **S** using 5.4 g of the host **1S**. We could successfully obtain 1.09 g of **P1**, showing that this polymerization system is indeed scalable. In addition to the scalability, the recyclability of **1S** was demonstrated after the decomposition of the framework. Crystalline **1S** was successfully reproduced from the remaining **S** and Cu by the addition of pyridine and deficient **S** in methanol solution. These data clearly showed the advantage of this system with regard to the scalability and recyclability. We added these descriptions in the revised version of the manuscript (page 8, lines 5-9, page 16, lines 15-16; Supplementary information page 4, lines 1-5).

Reviewer #3

Comment 1: Having looked at the revision, the authors have included substantial new experimental data. However, they still do not address the fundamental limitation of the present paper, which they also state in their comments: "... SAAA repetitive sequence is predominantly formed, ... ", but not exclusively.

As such, I would recommend to submit the manuscript to a more specialized journal.

Answer 1: In this work, the monomer sequences in our copolymers were carefully analyzed from multiple points of view. The result of copolymerization at different monomer feed ratios provided the crucial evidence for the existence of periodic **SAAA** monomer sequence in **P1**. For the detailed investigation on the monomer sequence, we performed NMR spectroscopies of copolymers, showing that the predominant triads in **P1** are **ASA**, **AAS**, and

AAA. Considering the crystal structure of **1S**, the formation of these triads seemed to be valid. In addition to these analyses, MD simulations and DFT calculations were performed to support the experimental results, showing that the generation of repetitive **SAAA** sequence during propagation reaction was quite reasonable. Actually, we cannot say in our manuscript that a copolymer with “perfect” **SAAA** repetitive sequence is obtained because every copolymer must have some error sequences whatever the methodology is. We believe that this work has significant importance not only on sequence controlled radical polymerization but also on the development of tailor-made synthesis of desired polymeric materials utilizing the periodic structures of PCPs.

REVIEWERS' COMMENTS:

Reviewer #1 (Remarks to the Author):

Taking account the last revisions provided by the authors, I believe that this paper is now publishable in Nature Communications.

The authors have taken into account the final remark of reviewer#3. It is clear that their sequences contain defects and are not purely SAAA. Thus, it is recommended to clearly point out in the text that the copolymerization leads to a sequence-regulated trend but not to sequence-specific polymers.

Reviewer #2 (Remarks to the Author):

The authors have addressed the comments. However, their update to the SI was half-hearted at best. SIs should be written so that others could repeat the experiments. Statements like, "the equivalent molar of pyridine again Cu ion was the dropped into..." are not specific enough. How much pyridine was added? How much additional S was added to realize an equimolar mixture of Cu and S? If the authors add a detailed step-by-step procedure, then I will strongly support publication of the revised manuscript.

RESPONSE TO REVIEWERS' COMMENTS:

Reviewer #1

Comment 1: The authors have taken into account the final remark of reviewer#3. It is clear that their sequences contain defects and are not purely SAAA. Thus, it is recommended to clearly point out in the text that the copolymerization leads to a sequence-regulated trend but not to sequence-specific polymers.

Answer 1: Thank you for your important suggestion. We agreed this opinion, thus we described in the revised version that the formation of repetitive SAAA was the “predominant” or “most likely” sequence in our copolymer (e.g. page 2, line 11 / page 11, line 9).

Reviewer #2

Comment 1: The authors have addressed the comments. However, their update to the SI was half-hearted at best. SIs should be written so that others could repeat the experiments. Statements like, "the equivalent molar of pyridine again Cu ion was the dropped into..." are not specific enough. How much pyridine was added? How much additional S was added to realize an equimolar mixture of Cu and S? If the authors add a detailed step-by-step procedure, then I will strongly support publication of the revised manuscript.

Answer 1: We appreciate for your comment. In this revision, we added the details of the experiment for recycling PCP in Supplementary Information (Supplementary Information page 2, lines 1-9).